# Exploration of influenza incidence prediction model based on meteorological factors in Lanzhou, China, 2014–2017

Meixia Du[1,2☯], Hai Zhu[1☯], Xiaochun Yin[1,3]*, Ting Ke[1], Yonge Gu[1,3], Sheng Li[4]*, Yongjun Li[5], Guisen Zheng[1,3]

**1** School of Public Health, Gansu University of Chinese Medicine, Gansu Lanzhou, China, **2** Gansu Provincial Cancer Hospital, Gansu Lanzhou, China, **3** The Collaborative Innovation Center for Prevention and Control by Chinese Medicine on Disease Related Northwestern Environment and Nutrition, Gansu Lanzhou, China, **4** First People's Hospital of Lanzhou City, Gansu Lanzhou, China, **5** Gansu Provincial Center for Disease Control and Prevention, Gansu Lanzhou, China

☯ These authors contributed equally to this work.
* lzyxc@126.com (XY); 1178708407@qq.com (SL)

**Data Availability Statement:** All relevant data is within the paper and Supporting information files.

**Funding:** This work was supported by Innovation Fund Project of Higher Education in Gansu Province (2021B-159), Open Foundation of

## Abstract

Humans are susceptible to influenza. The influenza virus spreads quickly and behave seasonally. The seasonality and spread of influenza are often associated with meteorological factors and have spatio-temporal differences. Based on the influenza cases and daily average meteorological factors in Lanzhou from 2014 to 2017, this study firstly aimed to analyze the characteristics of influenza incidence in Lanzhou and the impact of meteorological factors on influenza activities. Then, SARIMA(X) models for the prediction were established. The influenza cases in Lanzhou from 2014 to 2017 was more male than female, and the younger the age, the higher the susceptibility; the epidemic characteristics showed that there is a peak in winter, a secondary peak in spring, and a trough in summer and autumn. The influenza cases in Lanzhou increased with increasing daily pressure, decreasing precipitation, average relative humidity, hours of sunshine, average daily temperature and average daily wind speed. Low temperature was a significant driving factor for the increase of transmission intensity of seasonal influenza. The SARIMAX (1,0,0)(1,0,1)[12] multivariable model with average temperature has better prediction performance than the university model. This model is helpful to establish an early warning system, and provide important evidence for the development of influenza control policies and public health interventions.

## Introduction

Influenza is an acute infectious respiratory disease caused by influenza virus [1], which has strong ability to mutate and fast propagation speed. The population is generally susceptible to influenza virus, causing the epidemic or outbreak [2, 3]. Around 5%-10% of adults and 20%-30% of children worldwide fall ill with influenza each year, and the outbreaks of influenza can

Collaborative Innovation Center for Prevention and Control by Chinese Medicine on Disease Related Northwestern Environment and Nutrition (99860202) and Open Foundation of Traditional Chinese Medicine Research Center of Gansu Province (ZYZX-2020-ZX16), Talent introduction project of Gansu University of Chinese Medicine (2018YJRC-10). The funders had no role in study design, data collection and analysis, decision to publish, or preparation of the manuscript.

**Competing interests:** NO authors have competing interests.

cause 3 to 5 million cases of severe illness and 290,000 to 650,000 respiratory deaths each year [4, 5]. From 2016 to 2019, the incidence of influenza in China increased from 22.3727/100,000 to 253.3561/100,000, and the mortality rate increased from 0.0041/100,000 to 0.0193/100,000 [6], both morbidity and mortality were on the rise. The influenza epidemic has imposed a great economic and health burden on human beings [7–10]. For the outbreak of influenza, there are many factors (such as individual genetic differences, changing population demographics, antibiotic resistance and environmental, etc.) [11–14] can affect the spread and infection of flu, therefore, many researchers have explored and analyzed these factors to prevent and control the influenza in some degree.

In recent years, more and more studies have proposed the importance of spatiotemporal heterogeneity to influenza activity [15]. In the tropics, there was a high level of influenza activity throughout the year with no obvious seasonality [16, 17]. Interestingly, the effects of meteorological conditions on influenza activity in the tropics were complex. In Antananarivo, influenza activity was not affected by meteorological conditions [18]; in Okinawa (Japan), both ambient temperature and relative humidity were inversely associated with influenza infection [19]; and positive association with humidity was found in El Salvador and Panama [20]. However, at present, it is generally believed that the influenza epidemic in temperate regions has obvious seasonality, epidemic peak occurred in cold winter [21, 22]. And influenza is negatively correlated with relative humidity and temperature, such as Islamabad (Pakistan) [23], Toronto (Canada) [24], Kenya [25], Shaoyang (China) [26]. In addition, an animal experiment found that the aerosol spread of influenza virus depended on the relative humidity and temperature of the environment, cold and dry meteorological conditions were conducive to the transmission of influenza virus [27].

Lanzhou, located in the semi-arid region of northwest China, is a temperate continental climate with large temperature difference and little precipitation [28]. The annual average temperature in Lanzhou is 10.3℃, and the annual average precipitation is 327 mm, which is mainly concentrated in June to September [29]. Influenza in Lanzhou occurs seasonally, up to now, there are few studies on forecasting the cases of influenza per month. More, early-warning-based interventions are important for improving disease control, community-based interventions, and personal protection [30]. Thus, the prediction of the cases of influenza in Lanzhou can provide a possibility for preventing and controlling influenza transmission for the government.

With the gradual development of computing function, the prediction models are gradually diversified, SIR [31], SARIMA [32], recurrent neural network [33] and other prediction models have been successfully used to predict influenza activities. Among them, the SARIMA model is the most popular and classic time series forecasting model because of its simplicity, system structure, and acceptable prediction performance [34]. Since meteorological factors have a certain influence on influenza activities, many researches used meteorological factors as exogenous variables to construct multivariate SARIMAX models. The results showed that the SARIMAX model including environmental variables can improve the prediction ability of influenza [35, 36].

Therefore, based on the influenza incidence and daily meteorological data (daily average atmospheric pressure, precipitation, average relative humidity, hours of Sunshine, daily average temperature, maximum temperature, minimum temperature and daily average wind speed) in Lanzhou from January 2014 to December 2017, the characteristics of influenza incidence was analyzed and the impact of meteorological factors on influenza activities in Lanzhou was studied. The SARIMA(X) prediction model was fitted according to the characteristics of influenza incidence and related meteorological factors in Lanzhou, providing a theory reference for early prevention and control of influenza.

## Materials and methods

### Study area

Lanzhou lies in the middle of Gansu Province, in the west of Longxi Loess Plateau and the northeast edge of Qinghai-Tibet Plateau. Geographical position of Lanzhou is 35˚34'20"~37˚07'07"N, 102˚35'58"~104˚34'29"E. Lanzhou city is long and narrow from east to west due to the influence of the terrain by the north and south mountains, it is a typical valley city in arid area of northwest China [37, 38].

### Data

**Influenza incidence data.**   The data of influenza cases were obtained from the National Infectious Disease Reporting Information Management System from January 2014 to December 2017. The daily incidence data of influenza cases in Lanzhou were all "reviewed" cases (cases confirmed after laboratory etiological diagnosis of influenza-like cases reported from sentinel hospitals by Lanzhou Center for Disease Control and Prevention).

The influenza cases in this study are confirmed diagnostic cases, and the diagnostic criteria are as follows:

Clinical manifestations of influenza (mainly fever, headache, myalgia and onset of general discomfort, body temperature up to 39–40 ºC, accompanied by systemic muscle and joint pain, fatigue, loss of appetite and other systemic symptoms, often sore throat, dry cough), positive results of one or more of the followed etiological tests:

1. Positive nucleic acid test of influenza virus (real-time RT-PCR and RT-PCR methods can be used).

2. Influenza virus isolation and culture were positive.

3. The level of influenza virus specific IgG antibody in both acute and convalescent sera increased by 4 or more times.

**Meteorological data.**   Daily meteorological data of Lanzhou were provided by Gansu Meteorological Bureau, from January 2014 to December 2017. Meteorological factors include daily average atmospheric pressure, precipitation, average relative humidity, hours of Sunshine, daily average temperature and daily average wind speed.

**Data preprocessing.**   The "date of case onset" in the influenza case data was used as a single variable frequency count to obtain the number of cases on a daily basis from January 1, 2014 to December 31, 2017. If there was no case onset on a certain day in a certain year or month, the observed number of cases on that day was recorded as 0. For the missing meteorological data, the mean value of adjacent points was used to repair the data. Finally, the daily incidence data and daily meteorological data were matched one by one. Excel 2010 software was used to integrate the data, establish a database and form a time series file.

Because the daily reported influenza cases were low, the influenza cases and meteorological factors data should be converted to monthly data during the establishment of SARIMA(X) model: the influenza cases are the sum of influenza cases per month, and meteorological factors are the monthly average of daily data.

### Data analysis

**Correlation analysis.**   Spearman correlation analysis was used to explore the correlation between influenza incidence and meteorological factors in Lanzhou, $\alpha = 0.05$.

**SARIMA(X) model.** Autoregressive integrated moving average (ARIMA) model is a famous time series prediction method proposed by Box and Jenkins in the early 1970s [39]. Because of the seasonal nature of influenza activity, the seasonal autoregressive integrated moving average (SARIMA) model was used in this study. The SARIMA model is abbreviated as SARIMA $(p, d, q) \times (P, D, Q)$ s, where s represents the seasonal period, $p$ ($P$), $d$ ($D$), and $q$ ($Q$) non-seasonal (seasonal) autoregressive, difference and moving average order. The equation of the model is as follows:

$$\phi(B)\varphi(B^s)(1 - B)^D Y_t = \theta(B)\Theta(B^s)\Sigma_t$$

Where $Y_t$ is the predicted value of the time series model at time t, B the backshift operator, $\phi(B)$ the autoregressive operator, $\varphi(B^s)$ the seasonal autoregressive operator, and $\theta(B)$ the shift Average operator, $\Theta(B^s)$ the seasonal moving average operator, and $\Sigma_t$ the random error. As a rule, the standard statistical methodology to establish a SARIMA model includes three steps: identification, parameter estimation, and diagnostic checking.

1. Evaluate the stability of the series (if it was not stability, the differencing was applied until the stability is achieved). An augmented Dickey-Fuller (ADF) test could determine whether the time series after differencing was stationary.

2. Estimate the $p$ ($P$) and $q$ ($Q$) parameters based on the autocorrelation function (ACF) and the partial autocorrelation function (PACF) plots. Generally, more than one tentative model was selected in this step.

3. Finally, the adequacy of the established model for the series was verified using the Box-Jenkins Q test [40] to check whether the residuals were equivalent to white noise. Then, the best SARIMA model was chosen from the possible models using the Akaike Information Criterion (AIC), where the fitted model was the one with the lowest AIC value.

Based on the optimal SARIMA model, a multivariate SARIMA model including meteorological factors as external regressors is further developed, and is referred to as the SARIMAX model.

For software implementation and visualization, the programming language R was used in the R 3.6.3 software environment. The time series data were divided into two parts as follows: the first 36 months (2014/01-2016/12) were used to construct the time series model with the Box and Jenkins approach and the last year (2017/01-2017/12) was used to validate the predictive power of the model. The predictive effect of SARIMA and SARIMAX models on the influenza cases in Lanzhou were evaluated by the accuracy of prediction, Root Mean Squared Error (RMSE)、Mean Absolute Error (MAE) and Mean Absolute Percentage Error (MAPE).

## Results

### Influenza surveillance

A total of 2,930 cases of influenza were reported in Lanzhou from 2014 to 2017, with a male-to-female ratio of 1.17 (1,581:1,349), and the influenza cases per year in males were higher than in females. Influenza patients ranged in age from 1 month to 90 years old. All age groups are susceptible, but proportion of the age of 0–10 years old influenza patients is 55.63%. The majority of cases of influenza were concentrated in winter (December—February of the following year), accounting for 44% (Table 1).

**Table 1. Epidemic surveillance of influenza in Lanzhou from 2014 to 2017.**

| Variable | | 2014 (%) | 2015 (%) | 2016 (%) | 2017 (%) | Over (%) |
|---|---|---|---|---|---|---|
| Gender | Male | 325(54.53) | 176(50.29) | 654(53.52) | 426(55.91) | 1581(53.96) |
| | Female | 271(45.47) | 174(49.71) | 568(46.48) | 336(44.09) | 1349(46.04) |
| Age | 0~ | 145(24.33) | 58(16.57) | 348(28.48) | 229(30.05) | 780(26.62) |
| | 6~ | 106(17.79) | 34(9.71) | 537(43.94) | 173(22.70) | 850(29.01) |
| | 11~ | 34(5.70) | 25(7.14) | 73(5.97) | 55(7.22) | 187(6.38) |
| | 21~ | 71(11.91) | 64(18.29) | 105(8.59) | 97(12.73) | 337(11.50) |
| | 41~ | 146(24.50) | 110(31.43) | 101(8.27) | 112(14.70) | 469(16.01) |
| | >61 | 94(15.77) | 59(16.86) | 58(4.75) | 96(12.60) | 307(10.48) |
| Seasonal | Spring | 78(13.09) | 97(27.71) | 682(55.81) | 129(16.93) | 986(33.65) |
| | Summer | 60(10.07) | 47(13.43) | 29(2.37) | 29(3.81) | 165(5.63) |
| | Autumn | 93(15.60) | 81(23.14) | 88(7.20) | 228(29.92) | 490(16.72) |
| | Winter | 365(61.24) | 125(35.72) | 423(34.62) | 376(49.34) | 1289(44.00) |

## Influenza activity and meteorological factors

The trend changes of daily influenza cases and meteorological factors in Lanzhou from 2014 to 2017 were analyzed, as shown in Table 2 and Fig 1. The average influenza cases per day was 2.01, with a maximum of 54 cases per day. The daily average atmospheric pressure, precipitation, average relative humidity, daily average temperature and sunshine hours were 848.31pha, 0.91 mm, 50.78% 11.34°C and 6.23 h, and daily average wind speed was 1.16 m/s. Fig 1 visually showed the changes of influenza activities and meteorological factors in Lanzhou from 2014 to 2017. We observed a significant seasonal variation for both influenza and meteorological factors, there is also a link between influenza activity and changes in meteorological factors.

## Correlation between influenza activity and meteorological factors

Spearman correlation analysis was used to analyze the daily influenza cases and daily average meteorological factors in Lanzhou from 2014 to 2017. The matrix of Spearman correlation coefficients within the meteorological factors was showed in Table 3. There was a positive correlation between the influenza cases and daily average atmospheric pressure, and other factors (precipitation, average relative humidity, hours of Sunshine, daily average temperature and daily average wind speed) were negatively correlated. The correlation coefficient was -0.489 between the influenza cases and temperature, and it was -0.088 between the influenza cases and relative humidity. In addition, atmospheric pressure and precipitation, relative humidity and temperature and other meteorological factors also have certain

**Table 2. Descriptive statistical analysis of meteorological elements and influenza activity in Lanzhou City during 2014–2017.**

| | Mean | SD | Min | P25 | P50 | P75 | Max |
|---|---|---|---|---|---|---|---|
| Cases | 2.01 | 4.36 | 0.00 | 0.00 | 1.00 | 2.00 | 54.00 |
| P (pha) | 848.31 | 5.30 | 833.00 | 844.30 | 848.10 | 852.00 | 866.30 |
| pre (mm) | 0.91 | 3.38 | 0.00 | 0.00 | 0.00 | 0.00 | 44.50 |
| RH (%) | 50.78 | 15.12 | 17.00 | 39.00 | 50.00 | 61.00 | 96.00 |
| Sun (h) | 6.23 | 3.77 | 0.00 | 3.50 | 6.60 | 9.30 | 12.80 |
| T-mean (°C) | 11.34 | 9.79 | -12.30 | 2.40 | 12.70 | 19.70 | 30.80 |
| Wsd (m/s) | 1.16 | 0.36 | 0.00 | 0.90 | 1.10 | 1.40 | 2.90 |

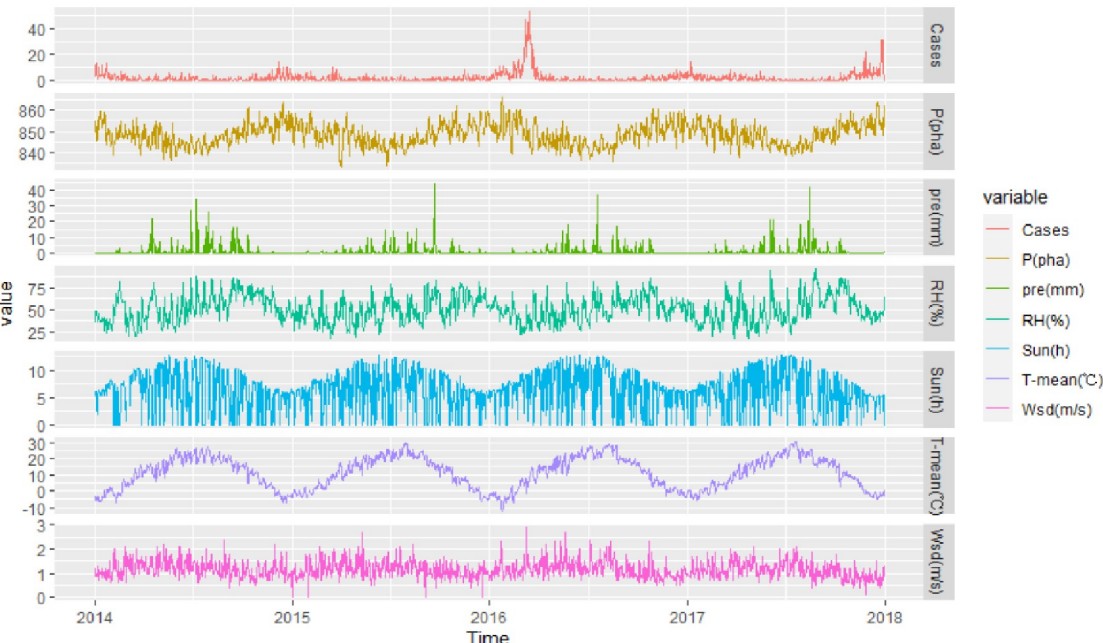

**Fig 1. The time series of daily meteorological factors and daily influenza cases in Lanzhou, 2014–2017 (Cases: Daily influenza cases, P (pha): Atmospheric pressure, pre (mm): Daily precipitation, RH (%): Relative humidity, Sun (h): Hours of sunshine, T-mean (˚C): Mean temperature, Wsd (m/s): Wind velocity).**

correlations, among which the correlation coefficient between atmospheric pressure and temperature is the highest ($r$ = -0.719, $P < 0.01$).

## Identifying the SARIMA(X) model

**Univariate ARIMA model.** The influenza cases in Lanzhou from January 2014 to December 2016 was taken as the training set, and its sequence was tested by Ljung-Box statistical test. The test showed that the original data sequence was white noise sequence ($P$ = 0.1554). Then ADF test was performed on the original sequence, the result showed that the influenza incidence sequence of the training set was stationary ($P < 0.001$). Therefore, it is not necessary to make a difference for this sequence in ARIMA($p$, $d$, $q$) ($P$, $D$, $Q$) $s$ model, where $d$ = 0, $D$ = 0

**Table 3. Spearman correlation results between daily meteorological variables and confirmed influenza cases in Lanzhou, China, 2014–2017.**

|  | Cases | P | pre | RH | Sun | T-mean | Wsd |
|---|---|---|---|---|---|---|---|
| Cases | 1 |  |  |  |  |  |  |
| P | 0.305** | 1 |  |  |  |  |  |
| Pre | -0.170** | -0.047 | 1 |  |  |  |  |
| RH | -0.088** | 0.141** | 0.501** | 1 |  |  |  |
| Sun | -0.110** | -0.312** | -0.340** | -0.415** | 1 |  |  |
| T-mean | -0.489** | -0.719** | 0.168** | -0.034 | 0.359** | 1 |  |
| Wsd | -0.169** | -0.250** | 0.081** | -0.185** | 0.092** | 0.253** | 1 |

$**P < 0.01$.

Cases: Daily influenza cases, P: Atmospheric pressure (pha), pre: daily precipitation (mm), RH: Relative humidity (%), Sun: hours of sunshine (h), T-mean: Mean temperature (˚C), Wsd: Wind velocity (m/s).

**Table 4. Selection of the univariate SARIMA model.**

| Model | RMSE | MAE | MAPE | AIC | BIC | $P°$ |
|---|---|---|---|---|---|---|
| (1,0,0)(1,0,1) [12] | 102.58 | 53.82 | 151.36 | 445.80 | 453.72 | 0.9588 |
| (0,0,1)(1,0,1) [12] | 103.00 | 53.62 | 152.30 | 446.08 | 454.00 | 0.8939 |
| (1,0,1)(1,0,1) [12] | 102.57 | 53.92 | 152.02 | 447.79 | 457.29 | 0.9760 |
| (1,0,0)(1,0,2) [12] | 102.58 | 53.83 | 151.46 | 447.8 | 457.3 | 0.9586 |
| (0,0,1)(1,0,2) [12] | 103.00 | 53.61 | 152.24 | 448.08 | 457.58 | 0.8943 |
| (1,0,1)(1,0,2) [12] | 102.94 | 55.06 | 166.87 | 450.85 | 461.94 | 0.1157 |

° Ljung-Box statistical test p-value, test level α = 0.05.

and seasonal period $s$ = 12. The values of parameters $p$ (P) and $q$ (Q) were determined according to ACF and PACF diagrams (S1 Fig). As shown in Table 4, six alternative univariate SARIMA models were constructed. The results of Ljung-Box test suggested that all the residual series of these models were white noise sequences. Based on the AIC and BIC, the best-fitting model was determined to be SARIMA (1,0,0)(1,0,1)[12], with a minimum AIC = 445.80 and a minimum BIC = 453.72.

**Multivariate SARIMAX model.** Meteorological factors such as P, pre, RH, Sun, T-mean and wsd were added individually or in combination to univariate ARIMA (1,0,0)(1,0,1)[12] as exogenous variables. A total of 56 SARIMAX models were fitted (S1 Table). These models were all statistically significant (Ljung-Box test $P$ > 0.05). SARIMA (1,0,0)(1,0,1)[12] with daily average temperature (T-mean) as external predictors was the optimal SARIMAX model, with a minimum AIC = 441.34 and a minimum BIC = 453.84, (RMSE = 98.05, MAE = 41.90, MAPE = 83.69, $P°$ = 0.9695).

**Model prediction and evaluation.** According to the optimal SARIMA(X) model, the influenza cases in Lanzhou was fitted from January 2017 to December 2017. As can be seen from Fig 2 and Table 5, the true values of the influenza cases were included in the 95% confidence interval of SARIMA model and SARIMAX model. The fitting trend of SARIMA + T-mean model was consistent with the real value, and with a relatively small RE(%) the fitting effect was good. RMSE, MAE and MAPE values were smaller than those of univariate SARIMA model.

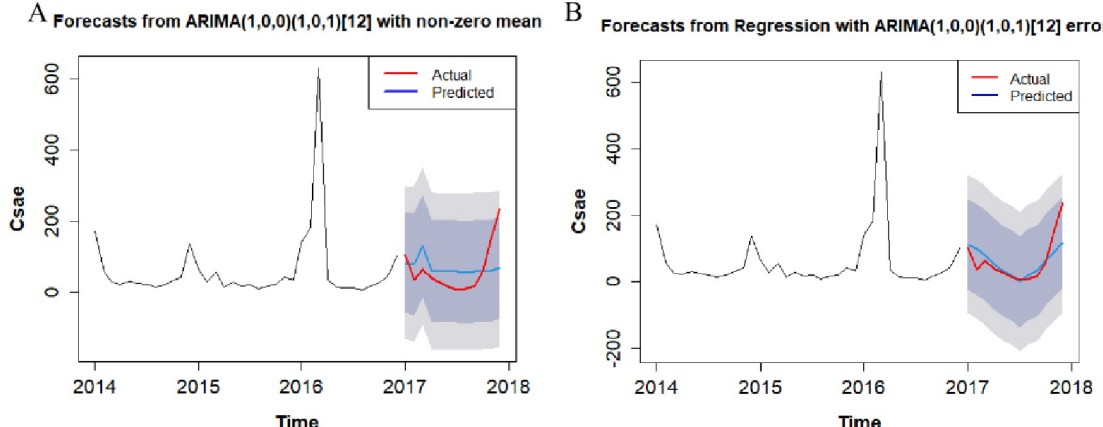

**Fig 2. Prediction fitting of SARIMA(X) model on the number of influenza cases in Lanzhou from January 2017 to December 2017 (A represents the Univariate SARIMA model, B represents the Multivariate SARIMAX model).**

**Table 5. Prediction results of SARIMA(X) model on the number of influenza cases in Lanzhou from January 2017 to December 2017.**

| Time | Actual | SARIMA(1,0,0)(1,0,1)[12] | | | | | | SARIMA(1,0,0)(1,0,1)[12] + T-mean | | | | | |
|------|--------|-----------|-------|--------|---------|--------|--------|-----------|--------|--------|---------|--------|--------|
| | | Predicted | Lo 80 | Hi 80 | Lo 95 | Hi 95 | RE(%) | Predicted | Lo 80 | Hi 80 | Lo 95 | Hi 95 | RE(%) |
| Jan. 2017 | 105 | 82.04 | -57.41 | 221.48 | -131.22 | 295.30 | -21.87 | 112.81 | -22.61 | 248.23 | -94.30 | 319.91 | 7.44 |
| Feb. 2017 | 35 | 79.43 | -64.10 | 222.87 | -139.94 | 298.81 | 126.94 | 97.43 | -39.24 | 234.11 | -111.60 | 306.46 | 178.37 |
| Mar. 2017 | 63 | 130.08 | -13.60 | 273.75 | -89.65 | 349.80 | 106.48 | 79.63 | -57.07 | 216.33 | -129.44 | 288.70 | 26.40 |
| Apr. 2017 | 39 | 60.10 | -83.59 | 203.78 | -159.65 | 279.84 | 54.10 | 53.44 | -83.27 | 190.14 | -155.63 | 262.50 | 37.03 |
| May. 2017 | 27 | 57.47 | -86.21 | 201.16 | -162.27 | 277.22 | 112.85 | 33.46 | -103.24 | 170.16 | -175.61 | 242.52 | 23.93 |
| Jun. 2017 | 15 | 57.02 | -86.66 | 200.71 | -162.72 | 276.77 | 280.13 | 19.97 | -116.74 | 156.67 | -189.10 | 229.03 | 33.13 |
| Jul. 2017 | 6 | 57.00 | -86.68 | 200.69 | -162.74 | 276.75 | 850.00 | 0.61 | -136.09 | 137.31 | -208.46 | 209.68 | -89.83 |
| Aug. 2017 | 8 | 56.57 | -87.11 | 200.26 | -163.17 | 276.32 | 607.13 | 20.94 | -115.76 | 157.64 | -188.12 | 230.01 | 161.75 |
| Sep. 2017 | 17 | 57.60 | -86.09 | 201.28 | -162.15 | 277.34 | 238.82 | 32.28 | -104.43 | 168.98 | -176.79 | 241.34 | 89.88 |
| Oct. 2017 | 55 | 58.99 | -84.69 | 202.68 | -160.76 | 278.74 | 7.25 | 63.50 | -73.20 | 200.20 | -145.57 | 272.57 | 15.45 |
| Nov. 2017 | 156 | 60.78 | -82.90 | 204.46 | -158.96 | 280.53 | -61.04 | 90.81 | -45.89 | 227.51 | -118.26 | 299.88 | -41.79 |
| Dec. 2017 | 236 | 67.50 | -76.18 | 211.19 | -152.24 | 287.25 | -71.40 | 116.64 | -20.06 | 253.34 | -92.43 | 325.71 | -50.58 |

## Discussion

Influenza is a common respiratory infection that causes considerable morbidity and mortality worldwide each year [41]. In recent years, with the development of databases and computing, there have been some important developments in influenza surveillance [42] and prediction, and the studies found that the SARIMAX model including environmental variables could improve the prediction ability [43]. In this paper, the influenza cases in Lanzhou from 2014 to 2017 were analyzed, the optimal prediction model was established, and whether adding meteorological factors as exogenous variables would have a better prediction effect was evaluated.

In this study, it was found that the majority of influenza cases in Lanzhou were male, and the incidence of influenza cases in the group less than 10 years old accounted for 55.63%. These researches were consistent with the research results in other areas, the younger the age, the higher the susceptibility [44–46]. People in the ≤ 10 years old group have low immunity, and spend most of their time in nursery institutions or schools and other collective units, they are the most vulnerable to infection which usually make influenza outbreaks or epidemics among them. Of the 12 influenza outbreaks in Liaoning province from 2010 to 2014, 10 occurred in nurseries and primary schools, accounting for 83.33% [47]. Influenza infection causes a significant burden of disease each year in the pediatric population worldwide [10, 48]. Immunizations against influenza is well tolerated and effective [49, 50], and flu vaccines can greatly reduce the disease burden. Jorge et al. found that when flu immunization programmes were expanded to cover Mexico's school-age population, there was an overall decrease in the economic burden for the Mexican health care system of 111.9 million US dollars [51]. Therefore, preferential vaccination policies can be implemented for younger age groups on the basis of conditions. At the same time, kindergartens, schools and other collective units need to take control measures to prevent influenza and other respiratory infections.

On the whole, influenza in China presents the epidemic characteristics showed that there is a peak in winter, a secondary peak in spring, and a trough in summer and autumn [52]. This characteristic is consistent with those studies in which that winter influenza epidemics caused considerable morbidity and mortality in temperate regions [53]. Cold and dry conditions were documented as major determinants of influenza seasonality in temperate regions, such as New

York, Berlin (Germany), Ljubljana (Slovenia), Castile and León (Spain) and Israeli District, Canada, etc. [54–58]. This is also consistent with the research results of this paper. Average daily temperature, precipitation and average relative humidity were negatively correlated with influenza activities due to longer survival time and higher survival rate of influenza virus in aerosol droplets at low temperature and relative humidity (between 20% and 35%), increasing the risk of airborne transmission [59]. On the contrary, temperature and relative humidity would impose an increase for the transmission of influenza virus in Yuhang District (China), northern Cameroon, Bangkok and other tropical areas [60–62]. Some studies suggest that this is because the wet conditions of tropical rainy seasons may encourage contact transmission of virus, by increasing the amount of virus that is deposited on surfaces, and by increasing virus survival in droplets on surfaces [63]. With cool-dry conditions enhancing influenza A virus survival and transmissibility in temperate climates in high latitudes, but humid-rainy conditions favor outbreaks in low latitudes [64]. The seemingly contradictory results further emphasize the complexity of the influence of climate factors and geographical environment on influenza activity. Air pressure was positively correlated with influenza activity, while influenza was negatively correlated with sunshine hours and daily mean wind speed, which was consistent with the results of Qi et al. [65] and previous research in Lanzhou from 2005 to 2010 [66]. When the air pressure is higher, the diffused virus in the air and is not easy to dissipate, resulting in an increase in influenza patients. However, for sunshine hours, some researchers suggest that decreased solar insolation during the winter months is posited to increase influenza activity by decreasing host melatonin and vitamin D levels and thus host resistance [67, 68]. Wind speed can affect the flow speed of air and reduce the concentration of virus in the air. The specific reason remains should to be further studied and analyzed.

Different climatic conditions and geographical environments have different impacts on influenza activities, so the characteristics and prediction models of influenza activities are regional. The established SARIMA model in this paper will benefit to predict the influenza cases in Lanzhou based, the correlations between the influenza cases and meteorological variables. This is similar to what the researchers have found elsewhere, when is that adding weather as an exogenous variable, the model was a better predictor of influenza activity. Chadsuthi et al. [69] studied the influenza prediction model in central and southern Thailand, and found that the ARIMAX model including the average temperature lagging 4 months and the minimum relative humidity lagging 2 months was the model suitable for the central region, while the ARIMAX model including the minimum relative humidity lagging 4 months was the best model suitable for the southern region. In Abidjan, the inclusion of rainfall can improve the performance of influenza fitting and forecasting models [70]. Although most studies have found that the inclusion of meteorological factors can improve the performance of influenza prediction models, due to the differences in regional climatic conditions and human factors, the meteorological factors included in the optimal multivariate models for influenza prediction in each region are different. This is a further reminder of the regional nature of influenza prediction models.

There are some limitations to the study. First, other factors such as economics and vaccination status that play a role in influencing the correlation between influenza activity and climate variables were not taken into account. Many studies have shown that vaccination is a cost-effective option for protecting vulnerable populations and reducing the incidence of influenza [71, 72]. Second, there is no (subtype) classification of influenza viruses. It is necessary to further subdivide the influenza viruses, and explore whether the forecasting models of different types of influenza viruses in Lanzhou are affected by different meteorological factors. A combination of time series analysis of influenza viruses and active epidemiological surveillance may help to more accurately predict future seasonal influenza activity.

## Conclusion

In Lanzhou, more male than female influenza cases occurred in 2014–2017, and the younger the age, the higher the susceptibility. Low temperature was a significant driving factor for the increase of seasonal influenza transmission intensity. carried out A seasonal autoregressive moving average (SARIMA) multivariate analysis of all influenza data from 2014 to 2017 was carried out and the SARIMAX (1,0,0)(1,0,1)[12] multivariate model based on average temperature has better prediction performance than the univariate model. This predictive model may help establish early warning systems and provide important evidence for influenza control policies and public health interventions.

## Supporting information

**S1 Fig. Autocorrelation (ACF) and Partial Autocorrelation (PACF) graphs of influenza data training sets.**
(DOCX)

**S1 Table. Meteorological factors were incorporated into the ARIMA(1,0,0)(1,0,1) [12]+X multivariable mode.**
(DOCX)

**S1 Data.**
(XLSX)

## Author Contributions

**Conceptualization:** Hai Zhu.

**Data curation:** Sheng Li, Yongjun Li.

**Funding acquisition:** Xiaochun Yin, Guisen Zheng.

**Methodology:** Meixia Du, Yonge Gu.

**Project administration:** Xiaochun Yin, Guisen Zheng.

**Software:** Meixia Du, Hai Zhu.

**Supervision:** Ting Ke, Sheng Li.

**Visualization:** Meixia Du.

**Writing – original draft:** Meixia Du, Hai Zhu, Ting Ke.

**Writing – review & editing:** Xiaochun Yin, Yonge Gu, Sheng Li, Yongjun Li, Guisen Zheng.

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
