## [Decision Letter · Decision Letter 0]

17 May 2022

PONE-D-21-37259Exploration of influenza incidence prediction model based on meteorological factors in Lanzhou, China, 2014-2017PLOS ONE

Dear Dr. Yin,

Thank you for submitting your manuscript to PLOS ONE. After careful consideration, we feel that it has merit but does not fully meet PLOS ONE’s publication criteria as it currently stands. Therefore, we invite you to submit a revised version of the manuscript that addresses the points raised during the review process.

ACADEMIC EDITOR: A comprehensive English editing is required.

We look forward to receiving your revised manuscript.

Kind regards,

Ka Chun Chong

Academic Editor

PLOS ONE

Journal Requirements:

“This work was supported by Innovation Fund Project of Higher Education in Gansu Province (2021B-159), Open Foundation of Collaborative Innovation Center for Prevention and Control by Chinese Medicine on Disease Related Northwestern Environment and Nutrition (99860202) and Open Foundation of Traditional Chinese Medicine Research Center of Gansu Province (ZYZX-2020-ZX16).”

“This work was supported by Innovation Fund Project of Higher Education in Gansu Province (2021B-159), Open Foundation of Collaborative Innovation Center for Prevention and Control by Chinese Medicine on Disease Related Northwestern Environment and Nutrition (99860202) and Open Foundation of Traditional Chinese Medicine Research Center of Gansu Province (ZYZX-2020-ZX16).Include this sentence at the end of your statement: The funders had no role in study design, data collection and analysis, decision to publish, or preparation of the manuscript.”

Reviewers' comments:

Reviewer's Responses to Questions

**Comments to the Author**

1. Is the manuscript technically sound, and do the data support the conclusions?

Reviewer #1: Yes

Reviewer #3: Yes

2. Has the statistical analysis been performed appropriately and rigorously? 

Reviewer #1: Yes

Reviewer #3: I Don't Know

3. Have the authors made all data underlying the findings in their manuscript fully available?

Reviewer #1: Yes

Reviewer #3: No

4. Is the manuscript presented in an intelligible fashion and written in standard English?

Reviewer #1: Yes

Reviewer #3: Yes

5. Review Comments to the Author

Reviewer #1: This manuscript compared the performances of the ARIMA model and ARIMAX model in predicting influenza cases in Lanzhou. The statistical analysis of this study is clearly described, and the results are nice. However, some issues need to be considered.

1. L105-107. Who was responsible for case report, only the public health organization? Or both public and private health agencies? How was the influenza case diagnosed (i.e. diagnostic criteria)?

2. L 107, What does the “reviewed” mean?

3. L136-138. It remains unclear that how the meteorological factors were included in the ARIMAX model. Did one model contain only one meteorological variable, or all variables were included in a model at the same time? The collinearity of meteorological factors should be considered if they were combined and included into models.

4. L140-143 “the first 36 months (2014/01-2016/12) were used to construct the time series model”. If the monthly data was used to construct the model, it should be clearly indicated that how the daily data was aggregated into monthly data.

5. L199. For the ADF test, P>0.001 could not indicate that the original sequence was stationary. The exact P-value should be given, and it should be compared with 0.05 to prove the stationary.

6. Fig 2 and Table 5 just provided the same information.

7. Please compare both the fitting and forecasting performance (e.g. RMSE) of these two models.

8. There was an influenza outbreak in 2016, while the other years showed slight fluctuation. This outbreak had a certain impact on the modelling performance. Thus, if there is no such outbreak, that is, the number of influenza cases changes slightly from year to year, does the ARIMAX model still have high performance? Since influenza outbreak is not common in Lanzhou, the above mentioned scenario should be considered.

9. How much data could the ARIMA/ARIMAX model predict? From Fig 2 and Table 5, the models showed relatively good predicting performance during the first half of 2017, but significant forecast error could be found in the second half of 2017. What might be the reasons?

10. Please discuss whether this ARIMAX model is applicable in other conditions (e.g. in other regions).

11. Please revise the language of this manuscript. And some typos need to be corrected.

Reviewer #3: 1. The description of daily incidence data could be more clear, as the “reviewed cases” has an ambiguous meaning.

2. Why the parameter of the seasonal period is set to be 12?

3. Maybe “Seasonal ARIMA” is a more accurate word to describe the model instead of ARIMA.

4. It would be better to clarify the formula of the seasonal ARIMA model in the method part.

5. Both the ACF and PACF plots do not have a significant spike, could you discuss this limitation, and does it matter to your model?

6. The AIC value did improve from ARIMA (445.80) and ARIMA(X) (441.34), but didn’t improve much, AIC and BIC indeed are the good methods to evaluate the model fit, but their values are larger when more variables were added. The optimal ARIMA(X) model is selected by the minimum AIC and BIC value, the procedure of comparing the models’ prediction error should also be displayed, which will make the optimal model more convincing.

6. PLOS authors have the option to publish the peer review history of their article (what does this mean?). If published, this will include your full peer review and any attached files.

Reviewer #1: No

Reviewer #3: No

---

## [Author Response · Author response to Decision Letter 0]

25 Aug 2022

Dear Editors and Reviewers:

 Hope you are doing well.

Firstly, thank you for your e-mail and the reviewers’ comments concerning about our manuscript entitled “Exploration of influenza incidence prediction model based on meteorological factors in Lanzhou, China, 2014-2017” (ID: PONE-D-21-37259). Those comments are all valuable and very helpful for revising and improving our paper, as well as the important guiding significance to our researches. Then, we have studied comments carefully and have made correction which we hope to meet with approval. Revised portion are marked in blue in the paper. The main corrections in the paper and the responds to the reviewer’s comments are as follows:

Respond to the Reviewer #1’s comments:

 1. L105-107. Who was responsible for case report, only the public health organization? Or both public and private health agencies? How was the influenza case diagnosed (i.e. diagnostic criteria)?

√ Response: 1. Responsibility reporter: The data used in this study were downloaded from the disease surveillance network and reported by sentinel hospitals (such as Gansu Provincial People's Hospital, Lanzhou University First Hospital, Lanzhou University Second Hospital, Chengguan District Hospital, etc.). 

2. Influenza sentinel surveillance agencies are mainly made up of public agencies, and no private agency is currently tasked with this task.

3. We have clarified the diagnostic criteria for influenza cases in line 116-127 of the article. 

Line 112-122: The influenza cases in this study are confirmed diagnostic cases, and the diagnostic criteria are:

Clinical manifestations of influenza (mainly fever, headache, myalgia and onset of general discomfort, body temperature up to 39-40 ºC, accompanied by systemic muscle and joint pain, fatigue, loss of appetite and other systemic symptoms, often sore throat, dry cough), positive results of one or more of the following etiological tests:

1. Positive nucleic acid test of influenza virus (real-time RT-PCR and RT-PCR methods can be used).

2. Influenza virus isolation and culture were positive.

3. The level of influenza virus specific IgG antibody in both acute and convalescent sera increased by 4 or more times.

 L 107, What does the “reviewed” mean?

√ Response: "Reviewed" refers to laboratory etiological diagnosis of influenza-like cases reported by sentinel hospitals by municipal Center for Disease Control and Prevention, and positive cases are confirmed cases of influenza. We have made a corresponding supplement in line 110-112 of the article.

Line 109-111: (cases confirmed after laboratory etiological diagnosis of influenza-like cases reported from sentinel hospitals by Lanzhou Center for Disease Control and Prevention).

 L136-138. It remains unclear that how the meteorological factors were included in the ARIMAX model. Did one model contain only one meteorological variable, or all variables were included in a model at the same time? The collinearity of meteorological factors should be considered if they were combined and included into models.

√ Response: We are very sorry for not showing the results of the 56 ARIMAX models fitted. In the establishment of ARIMAX models, meteorological factors were successively incorporated into the model. Compare the fit and prediction effects of ARIMAX models under different combinations of meteorological factors. We put together these 56 different models into Schedule 1.

Schedule 1 Meteorological factors were incorporated into the ARIMA(1,0,0)(1,0,1)[12]+X multivariable mode

 Model RMSE MAE MAPE AIC BIC P。

1 ARIMA + P 100.68 46.91 116.13 446.34 455.84 0.9789

2 ARIMA + T 98.05 41.90 83.69 441.34 453.84 0.9695

3 ARIMA + RH 94.44 53.88 173.02 444.99 454.5 0.9538

4 ARIMA + pre 93.37 51.85 152.57 445.01 454.51 0.9797

5 ARIMA + Wsd 102.45 52.63 142.19 447.67 457.17 0.9516

6 ARIMA + Sun 102.51 53.93 154.19 447.77 457.27 0.9543

7 ARIMA + P + T 94.32 39.98 76.71 445.79 456.87 0.9575

8 ARIMA + P + RH 78.00 41.98 139.11 441.44 452.53 0.9919

9 ARIMA + P + pre 94.28 49.63 139.19 446.83 457.92 0.9779

10 ARIMA + P + Wsd 96.32 46.55 131.67 447.62 458.7 0.9920

11 ARIMA + P + Sun 93.53 44.88 133.93 446.96 458.04 0.9912

12 ARIMA + T + RH 81.19 40.76 121.82 441.69 452.78 0.9994

13 ARIMA + T + Pre 94.63 42.83 92.81 445.87 456.95 0.9886

14 ARIMA + T + Wsd 88.59 39.78 94.61 444.81 455.89 0.9860

15 ARIMA + T + Sun 84.94 37.86 81.35 577.08 588.3 0.9788

16 ARIMA + RH + pre 85.04 49.15 166.94 444.52 455.59 0.9958

17 ARIMA + RH + Wsd 83.04 50.02 156.71 445.51 455.79 0.9858

18 ARIMA + RH + Sun 87.26 49.22 158.90 444.33 455.42 0.9746

19 ARIMA + pre + Wsd 92.95 52.52 158.12 446.98 458.06 0.9829

20 ARIMA + pre + Sun 89.71 52.31 171.47 446.46 457.54 0.9931

21 ARIMA + Wsd+ Sun 84.94 37.87 81.36 577.08 588.30 0.9788

22 ARIMA + P + T + RH 84.94 37.87 81.36 577.08 588.30 0.9788

23 ARIMA + P + T + pre 94.75 43.04 108.32 446.17 457.28 0.9949

24 ARIMA + P + T + Wsd 92.46 40.68 86.59 447.24 459.91 0.9449

25 ARIMA + P + T + Sun 88.89 41.82 119.24 445.67 458.34 0.9849

26 ARIMA + P + RH + pre 77.99 41.96 138.87 443.44 456.11 0.9918

27 ARIMA + P + RH + Wsd 77.81 41.63 136.47 443.40 456.07 0.9965

28 ARIMA + P + RH + Sun 77.90 42.03 138.95 443.45 456.11 0.9909

29 ARIMA + P + pre + Wsd 90.48 48.71 149.62 448.16 460.83 0.9900

30 ARIMA + P + pre + Sun 85.02 47.36 157.73 446.78 459.44 0.9936

31 ARIMA + P + Wsd + Sun 90.47 46.13 145.88 448.30 460.97 0.9714

32 ARIMA + T + RH + pre 81.26 40.84 122.13 443.69 456.36 0.9994

33 ARIMA + T + RH + Wsd 85.26 39.14 118.25 454.97 466.12 0.9984

34 ARIMA + T + RH + Sun 81.42 40.42 119.35 443.63 456.3 0.9974

35 ARIMA + T + pre + Wsd 87.68 42.62 112.13 446.47 459.14 0.9938

36 ARIMA + T + pre + Sun 84.94 37.87 81.36 577.08 588.30 0.9788

37 ARIMA + T + Wsd+ Sun 84.94 37.87 81.36 577.08 588.30 0.9788

38 ARIMA + RH + pre + Wsd 81.29 37.04 124.71 444.03 491.03 0.9198

39 ARIMA + RH + pre + Sun 84.41 48.58 162.27 445.83 458.50 0.9870

40 ARIMA + RH + Wsd + Sun 84.41 48.58 162.27 445.83 458.50 0.9870

41 ARIMA + pre + Wsd + Sun 89.01 51.12 166.74 448.28 460.95 0.9915

42 ARIMA + P + T + RH + pre 94.24 42.87 112.12 455.5 465.07 0.9711

43 ARIMA + P + T + RH + Wsd 89.46 46.32 151.45 447.75 452.32 0.9232

44 ARIMA + P + T + RH + Sun 89.01 51.12 166.74 448.28 460.95 0.9915

45 ARIMA + P + RH + pre + Wsd 77.84 41.63 136.28 445.40 459.65 0.9963

46 ARIMA + P + RH + pre + Sun 77.94 42.07 139.16 445.44 459.69 0.9920

47 ARIMA + P + pre + Wsd + Sun 84.37 46.46 156.32 448.54 462.79 0.9850

48 ARIMA + T + RH + pre + Wsd 84.94 37.86 81.35 577.08 588.3 0.9788

49 ARIMA + T + RH + pre + Sun 84.37 46.46 156.32 448.54 462.79 0.9850

50 ARIMA + T + pre + Wsd + Sun 82.95 41.60 123.38 446.52 460.77 0.9887

51 ARIMA + RH + pre + Wsd + Sun 82.95 41.60 123.38 446.52 460.77 0.9887

52 ARIMA + P + T + RH + pre + Wsd 78.62 40.45 129.00 447.26 463.1 0.9943

53 ARIMA + P + T + RH + pre + Sun 82.95 41.60 123.38 446.52 460.77 0.9887

54 ARIMA + P + RH + pre + Wsd + Sun 82.95 41.60 123.38 446.52 460.77 0.9887

55 ARIMA + T + RH + pre + Wsd + Sun 82.95 41.60 123.38 446.52 460.77 0.9887

56 ARIMA + P + T + RH + pre + Wsd + Sun 89.94 41.69 113.69 451.21 468.63 0.9888

 L140-143 “the first 36 months (2014/01-2016/12) were used to construct the time series model”. If the monthly data was used to construct the model, it should be clearly indicated that how the daily data was aggregated into monthly data.

√ Response: We are very sorry for not being clear about this area. The processing of data was supplemented on lines 129-141.

Line 128-140: 

Data preprocessing

The "date of case onset" in the influenza case data was used as a single variable frequency count to obtain the number of cases on a daily basis from January 1, 2014 to December 31, 2017. If there was no case onset on a certain day in a certain year or month, the observed number of cases on that day was recorded as 0. For the missing meteorological data, the mean value of adjacent points was used to repair the data. Finally, the daily incidence data and daily meteorological data were matched one by one. Excel 2010 software was used to integrate the data, establish a database and form a time series file.

Because of the low number of influenza cases reported daily, the influenza cases and meteorological factors data should be converted to monthly data during the establishment of SARIMA(X) model: the influenza cases are the sum of influenza cases per month, and meteorological factors are the monthly average of daily data.

 L199. For the ADF test, P>0.001 could not indicate that the original sequence was stationary. The exact P-value should be given, and it should be compared with 0.05 to prove the stationary.

√ Response: We are very sorry for the mistake in our sentence and “P > 0.001” have been changed to " P < 0. 001" in line 233 of the text.

 Response to comment: Fig 2 and Table 5 just provided the same information. 

√ Response: Yes, Figure 2 and Table 5 showed the same information, but they have different effects. Figure 2 is to compare the fitting effects of the two models more intuitively, and Table 5 is to give the exact fitting values of the two models.

 Please compare both the fitting and forecasting performance (e.g. RMSE) of these two models.

√ Response: The fitting and forecasting effects of the two models have been compared in original text. The univariate models were shown in Table 4, and the multivariate model was shown in lines 250-251.

Univariate models: 

Table 4 Selection of the univariate ARIMA model

Model RMSE MAE MAPE AIC BIC P。

(1,0,0)(1,0,1)[12] 102.58 53.82 151.36 445.80 453.72 0.9588

(0,0,1)(1,0,1) [12] 103.00 53.62 152.30 446.08 454.00 0.8939

(1,0,1)(1,0,1)[12] 102.57 53.92 152.02 447.79 457.29 0.9760

(1,0,0)(1,0,2)[12] 102.58 53.83 151.46 447.8 457.3 0.9586

(0,0,1)(1,0,2) [12] 103.00 53.61 152.24 448.08 457.58 0.8943

(1,0,1)(1,0,2)[12] 102.94 55.06 166.87 450.85 461.94 0.1157

Multivariate model: 

Lines 250-251: ARIMA (1,0,0)(1,0,1)[12] with daily average temperature (T-mean) as external predictors was the optimal ARIMAX model, with a minimum AIC = 441.34 and a minimum BIC = 453.84, (RMSE = 98.05, MAE = 41.90, MAPE = 83.69, P = 0.9695). 

 There was an influenza outbreak in 2016, while the other years showed slight fluctuation. This outbreak had a certain impact on the modelling performance. Thus, if there is no such outbreak, that is, the number of influenza cases changes slightly from year to year, does the ARIMAX model still have high performance? Since influenza outbreak is not common in Lanzhou, the abovementioned scenario should be considered.

√ Response: We took the average number of influenza cases in 2014 and 2015 as influenza cases from January to December in 2016 to reduce the fluctuation of data and fit the SARIMA(1,0,0)(1,0,1)[12] + T-mean model. The fitting results are shown in the figure below. The results showed that the fitting effect was worse, indicating that the fluctuation in 2016 had little impact on the prediction model, because the reported cases of influenza showed an upward trend from 2014 to 2017, and it was normal to have fluctuations from year to year.

 How much data could the ARIMA/ARIMAX model predict? From Fig 2 and Table 5, the models showed relatively good predicting performance during the first half of 2017, but significant forecast error could be found in the second half of 2017. What might be the reasons? 

√ Response: In this study, we used 3 years of data for the fitting model, which can predict 12 months, and its prediction from January to October was better. The reason for the poor effect of late prediction is that the number of influenza cases in winter of 2017 is higher than that of previous years, which leads to the low prediction result of the model.

 Please discuss whether this ARIMAX model is applicable in other conditions (e.g. in other regions).

√ Response: We have discussed in line 323-327 whether the ARIMAX model is applicable in other regions. 

For other parts of Gansu Province that are similar to Lanzhou's geographical environment, meteorological factors, economic conditions and other factors, we can also try to use this model to predict local influenza cases. For areas with large differences from Lanzhou (such as Beijing, Shanghai, etc.), we should explore appropriate ARIMAX models based on the influencing factors of influenza activity in other regions to predict local influenza activity.

 Please revise the language of this manuscript. And some typos need to be corrected.

√ Response: We feel very sorry for some spelling mistakes. We have checked and revised the full text and marked it in blue font.

Respond to the Reviewer #3’s comments:

 The description of daily incidence data could be clearer, as the “reviewed cases” has an ambiguous meaning.

√ Response: "Reviewed cases " refers to laboratory etiological diagnosis of influenza-like cases reported by sentinel hospitals by municipal Center for Disease Control and Prevention, and positive cases are confirmed cases of influenza. We have made a corresponding supplement in line 110-112 of the article.

Line 109-111: (cases confirmed after laboratory etiological diagnosis of influenza-like cases reported from sentinel hospitals by Lanzhou Center for Disease Control and Prevention).

 Why the parameter of the seasonal period is set to be 12?

√ Response: Because we used monthly data for forecast fitting, the period is 12 months.

 Maybe “Seasonal ARIMA” is a more accurate word to describe the model instead of ARIMA.

√ Response: Thank you very much for your suggestion. In this paper, we actually used the seasonal ARIMA model. We have changed ARIMA(X) to SARIMA(X) as shown in the blue font.

 It would be better to clarify the formula of the seasonal ARIMA model in the method part.

√ Response: We feel very sorry for not giving the formula of ARIMA model, which we added in line 152-157 of the article. 

Line 152-157: The equation of the model is as follows:

ϕ(B)φ(B^s ) (1-B)^D Y_t=θ(B)Θ(B^s ) Σ_t

Where〖 Y〗_t is the predicted value of the time series model at time t, B is the backshift operator, ϕ(B) is the autoregressive operator, φ(B^s ) is the seasonal autoregressive operator, and θ(B) is the shift Average operator, Θ(B^s ) is the seasonal moving averag e operator, and Σ_t is the random error.

 Both the ACF and PACF plots do not have a significant spike, could you discuss this limitation, and does it matter to your model? 

√ Response: ACF and PACF plots are used to determine model parameters and whether they are seasonal. Both ACF and PACF values within two standard deviations have little effect on our model. By reviewing the literature, we found that the ACF and PACF plots of the original sequences of certain variables in which Irie[1], Ilie[2] and Dou[3] were all within two standard deviations when establishing ARIMA model. 

[1] Ilie OD, Cojocariu RO, Ciobica A, Timofte SI, Mavroudis I, Doroftei B. 2020. Forecasting the Spreading of COVID-19 across Nine Countries from Europe, Asia, and the American Continents Using the ARIMA Models. Microorganisms. 30;8(8):1158. doi: 10.3390/microorganisms8081158. PMID: 32751609; PMCID: PMC7463904.

[2] Ilie OD, Ciobica A, Doroftei B. 2020. Testing the Accuracy of the ARIMA Models in Forecasting the Spreading of COVID-19 and the Associated Mortality Rate. Medicina (Kaunas). 56(11):566. doi: 10.3390/medicina56110566. PMID: 33121072; PMCID: PMC7694177. 

[3] Dou ZW, Ji MX, Wang M, Shao YN. 2022. Price Prediction of Pu'er tea based on ARIMA and BP Models. Neural Comput Appl. 34(5):3495-3511. doi: 10.1007/s00521-021-05827-9. Epub 2021 Mar 16. PMID: 33746365; PMCID: PMC7960402.

 The AIC value did improve from ARIMA (445.80) and ARIMA(X) (441.34), but didn’t improve much, AIC and BIC indeed are the good methods to evaluate the model fit, but their values are larger when more variables were added. The optimal ARIMA(X) model is selected by the minimum AIC and BIC value, the procedure of comparing the models’ prediction error should also be displayed, which will make the optimal model more convincing.

√ Response: Thanks for your comment. In the modeling process of this paper, we chose the optimal model according to the minimum principle of AIC and BIC. The results show that the fitting effect of the multivariate ARIMAX model established by adding the average temperature as an exogenous variable is better than that of the univariate ARIMA model, which also makes the model more convincing. 

Other changes:

 Line 3-4, We added an author, Sheng Li, who has made outstanding contributions to this paper, and changed him to the second corresponding author. Author Yongjun Li was replaced as the fifth author.

 Line 10, The second corresponding author's organization was added, “3 Lanzhou First People's Hospital, Gansu Lanzhou 730000, China”.

 Line 16, Added the address of the second corresponding author, “#b Lanzhou First People's Hospital, Gansu Lanzhou, China”

 Line 19, The second corresponding author's e-mail was added, “1178708407@qq.com”.

 Line 43-44, Modified statement “which has strong ability to mutate and fast propagation speed. The population is generally susceptible to influenza virus, causing the epidemic or outbreak”.

 Line 46, the statement of “influenza outbreaks cause” was corrected as “the outbreaks of influenza can cause”.

 Line 51-55, “Therefore, many researchers have explored and analyzed the factors that influenced the spread and infection of influenza，including individual genetic differences, changing population demographics, antibiotic resistance and environmental, etc.[11-14].” has been changed to “For the outbreak of influenza, there are many factors (such as individual genetic differences, changing population demographics, antibiotic resistance and environmental, etc.) [11-14] can affect the spread and infection of flu, therefore, many researchers have explored and analyzed these factors to prevent and control the influenza in some degree.”

 Line 63, “At present, however” has been changed to “However, at present”.

 Line 74-76, the statements of “Influenza in Lanzhou occurs seasonally, but there is no study on the prediction of the number of cases per month” were corrected as “Influenza in Lanzhou occurs seasonally, up to now, there are few studies on forecasting the cases of influenza per month.”.

 Line 77-79, “Thus, the prediction of the cases of influenza in Lanzhou can provide a possibility for preventing and controlling influenza transmission for the government.” was added.

 Line 93-94, “this study analyzed the characteristics of influenza incidence, and study the impact of meteorological factors on influenza activities in Lanzhou” has been changed to “the characteristics of influenza incidence was analyzed and the impact of meteorological factors on influenza activities in Lanzhou was studied”.

 Line 96-97, “providing a theory reference for early prevention and control of influenza.” was added.

 Line 147-149, “Because of the seasonal nature of influenza activity, the seasonal autoregressive integrated moving average (SARIMA) model was used in this study.” was added.

 Line 211-213, the statement of “Table 3 provides the matrix of Pearson’s correlation coefficients within the meteorological factors” has been changed to “The matrix of Spearman correlation coefficients within the meteorological factors was showed in Table 3”.

 Line 216-218, the statement of “The correlation coefficient was -0.489 between the influenza cases and temperature, and it was -0.088 between the influenza cases and relative humidity” was corrected as “The correlation coefficient between the number of influenza cases and temperature was -0.489, that between the number of influenza cases and relative humidity was -0.088.”.

 Line 220, “r” has been changed to “r”.

 Line 245, “daily average atmospheric pressure (P), precipitation (pre), average relative humidity (RH), hours of Sunshine (Sun), daily average temperature (T-mean) and daily average wind speed (wsd)” has been changed to “P, pre, RH, Sun, T-mean and wsd”.

 Line 255, “was” has been changed to “were”.

 Line 270-273, “This study mainly analyzed the number of influenza cases in Lanzhou from 2014 to 2017, explored the optimal prediction model, and evaluated whether adding meteorological factors as exogenous variables would have a better prediction effect” has been changed to “In this paper, the influenza cases in Lanzhou from 2014 to 2017 were analyzed, the optimal prediction model was established, and whether adding meteorological factors as exogenous variables would have a better prediction effect was evaluated”.

 Line 294-295, the statements of “There were also winter peak and spring secondary peak in the epidemic of influenza in Lanzhou. Studies have found that winter influenza epidemics caused considerable morbidity and mortality in temperate regions” were corrected as “This characteristic is consistent with those studies in which that winter influenza epidemics caused considerable morbidity and mortality in temperate regions”.

 Line 303-304, the statement of “there was a positive correlation between influenza activity and temperature and relative humidity” was corrected as “temperature and relative humidity would impose an increase for the transmission of influenza virus”.

 Line 316, “the virus diffuses” has been changed to “the diffused virus”.

 Line 324-328, the statements of “Using ARIMA model, the relationship between the number of influenza cases and meteorological variables makes the prediction model have significant predictive power. This is similar to what the researchers have found elsewhere, which is that adding weather as an exogenous variable was a better predictor of influenza activity” were corrected as “The established SARIMA model in this paper will benefit to predict the influenza cases in Lanzhou based, the correlations between the influenza cases and meteorological variables. This is similar to what the researchers have found elsewhere, when is that adding weather as an exogenous variable, the model was a better predictor of influenza activity”.

 Line 329-338, We put “Due to limitations in available and reliable data of influenza time-series in Lanzhou, the influenza forecasting accuracy is impaired. Therefore, we should continue to strengthen the epidemiological surveillance of influenza, as far as possible to reduce the failure rate. In addition, there may be differences in viral virulence that have not been considered” the limitations of article. 

 Line 379-383, we revised the funds.

Special thanks to you for your good comments. 

We tried our best to improve the manuscript and made some changes in the manuscript. These changes will not influence the content and framework of the paper. And here we did not list the changes but marked in blue in revised paper.

We appreciate for Editors and Reviewers’ warm work earnestly, and hope that the correction will meet with approval.

Once again, thank you very much for your comments and suggestions.

Best regards,

Dr.Xiaochun Yin

School of Public Health

Gansu University of Chinese Medicine

---

## [Decision Letter · Decision Letter 1]

12 Sep 2022

PONE-D-21-37259R1Exploration of influenza incidence prediction model based on meteorological factors in Lanzhou, China, 2014-2017PLOS ONE

Dear Dr. Yin,

Thank you for submitting your manuscript to PLOS ONE. After careful consideration, we feel that it has merit but does not fully meet PLOS ONE’s publication criteria as it currently stands. Therefore, we invite you to submit a revised version of the manuscript that addresses the points raised during the review process.

ACADEMIC EDITOR: Please address the remaining comments.

We look forward to receiving your revised manuscript.

Kind regards,

Ka Chun Chong

Academic Editor

PLOS ONE

Journal Requirements:

Reviewers' comments:

Reviewer's Responses to Questions

**Comments to the Author**

1. If the authors have adequately addressed your comments raised in a previous round of review and you feel that this manuscript is now acceptable for publication, you may indicate that here to bypass the “Comments to the Author” section, enter your conflict of interest statement in the “Confidential to Editor” section, and submit your "Accept" recommendation.

Reviewer #1: All comments have been addressed

Reviewer #2: (No Response)

Reviewer #3: All comments have been addressed

2. Is the manuscript technically sound, and do the data support the conclusions?

Reviewer #1: Yes

Reviewer #2: Partly

Reviewer #3: Yes

3. Has the statistical analysis been performed appropriately and rigorously? 

Reviewer #1: Yes

Reviewer #2: I Don't Know

Reviewer #3: Yes

4. Have the authors made all data underlying the findings in their manuscript fully available?

Reviewer #1: Yes

Reviewer #2: No

Reviewer #3: No

5. Is the manuscript presented in an intelligible fashion and written in standard English?

Reviewer #1: Yes

Reviewer #2: No

Reviewer #3: Yes

6. Review Comments to the Author

Reviewer #1: Some issues should still be considered, and minor revision is needed.

1. Please show both the fitting performance and the forecasting performance of both the optimal univariate model and the optimal multi-variate model. The fitting RMSE was calculated by compare the model fitted 2014-2016 data with the actual data in the same time period, while the forecasting RMSE was calculated by compare the predicted and actual data in 2017. Maybe the RMSEs in the Table 4 and Line 205 were the fitting performance? Table 5 showed the results of forecasted 2017 disease incidence, and the RMSE of each month should be shown in this table to indicate the forecasting performance. Also, in Table 5, the model output was the “Predicted” values instead of the “Fitted” values.

2. Line 325-327. Results of this study were compared with other relevant research, and it would be better to show more about the consistency and inconsistency compared with the similar research that conducted in the same/different regions, and the possible reasons for the difference.

3. Why the plots in Fig 2 had two titles? Please remove one if possible.

4. Some typos: e.g. Line176: 2016/012; Line 210: “Spearman line correlation”; Line 221: “。”; Table 4 please show the values with same number of decimals.

Reviewer #2: (No Response)

Reviewer #3: (No Response)

7. PLOS authors have the option to publish the peer review history of their article (what does this mean?). If published, this will include your full peer review and any attached files.

Reviewer #1: No

Reviewer #2: No

Reviewer #3: No

---

## [Author Response · Author response to Decision Letter 1]

15 Oct 2022

Dear Editors and Reviewers:

 Hope you are doing well.

Firstly, thank you for your e-mail and the reviewers’ comments concerning about our manuscript entitled “Exploration of influenza incidence prediction model based on meteorological factors in Lanzhou, China, 2014-2017” (ID: PONE-D-21-37259). Those comments are all valuable and very helpful for revising and improving our paper, as well as the important guiding significance to our researches. Then, we have studied comments carefully and have made correction which we hope to meet with approval. Revised portion are marked in blue in the paper. The main corrections in the paper and the responds to the reviewer’s comments are as follows:

Respond to the Reviewer #1’s comments:

 Please show both the fitting performance and the forecasting performance of both the optimal univariate model and the optimal multi-variate model. The fitting RMSE was calculated by compare the model fitted 2014-2016 data with the actual data in the same time period, while the forecasting RMSE was calculated by compare the predicted and actual data in 2017. Maybe the RMSEs in the Table 4 and Line 205 were the fitting performance? Table 5 showed the results of forecasted 2017 disease incidence, and the RMSE of each month should be shown in this table to indicate the forecasting performance. Also, in Table 5, the model output was the “Predicted” values instead of the “Fitted” values.

√ Response: Yes, the RMSEs, MAEs and MAPEs in Table 4 and line 205 were the fitting performance and the forecasting performance of the univariate models and the optimal multi-variate. 

RMSE is used to measure the discreteness of a group of numbers themselves. Suppose there were n training data x_i, the true output of each training data x_i was y_i, and the predicted value of x_i by the model was (y ) ^i. The RMSE of the model under n training data can be defined as follows [1]:

Therefore, RMSE is the evaluation index of model effect calculated according to each fixed ARIMA(X) model, and not every predicted value will have a corresponding RMSE value. In order to reflect the forecast effect, we also calculate the relative error of each month in Table 5.

In Table 5 and Figure 2, we have modified the “Fitted” values to the “Predicted” values.

 Guo Y, Feng Y, Qu F, Zhang L, Yan B, Lv J. Prediction of hepatitis E using machine learning models. PLoS One. 2020 Sep 17;15(9): e0237750. doi: 10.1371/journal.pone.0237750. 

Table 5 Prediction results of SARIMA(X) model on the number of influenza cases in Lanzhou from January 2017 to December 2017

Time Actual SARIMA(1,0,0)(1,0,1)[12] SARIMA(1,0,0)(1,0,1)[12] + T-mean

 Predicted Lo 80 Hi 80 Lo 95 Hi 95 RE(%) Predicted Lo 80 Hi 80 Lo 95 Hi 95 RE(%)

Jan. 2017

105 82.04 -57.41 221.48 -131.22 295.30 -21.87 112.81 -22.61 248.23 -94.30 319.91 7.44

Feb. 2017 35 79.43 -64.10 222.87 -139.94 298.81 126.94 97.43 -39.24 234.11 -111.60 306.46 178.37

Mar. 2017 63 130.08 -13.60 273.75 -89.65 349.80 106.48 79.63 -57.07 216.33 -129.44 288.70 26.40

Apr. 2017 39 60.10 -83.59 203.78 -159.65 279.84 54.10 53.44 -83.27 190.14 -155.63 262.50 37.03

May. 2017 27 57.47 -86.21 201.16 -162.27 277.22 112.85 33.46 -103.24 170.16 -175.61 242.52 23.93

Jun. 2017 15 57.02 -86.66 200.71 -162.72 276.77 280.13 19.97 -116.74 156.67 -189.10 229.03 33.13

Jul. 2017 6 57.00 -86.68 200.69 -162.74 276.75 850.00 0.61 -136.09 137.31 -208.46 209.68 -89.83

Aug. 2017 8 56.57 -87.11 200.26 -163.17 276.32 607.13 20.94 -115.76 157.64 -188.12 230.01 161.75

Sep. 2017 17 57.60 -86.09 201.28 -162.15 277.34 238.82 32.28 -104.43 168.98 -176.79 241.34 89.88

Oct. 2017 55 58.99 -84.69 202.68 -160.76 278.74 7.25 63.50 -73.20 200.20 -145.57 272.57 15.45

Nov. 2017 156 60.78 -82.90 204.46 -158.96 280.53 -61.04 90.81 -45.89 227.51 -118.26 299.88 -41.79

Dec. 2017 236 67.50 -76.18 211.19 -152.24 287.25 -71.40 116.64 -20.06 253.34 -92.43 325.71 -50.58

 Line 325-327. Results of this study were compared with other relevant research, and it would be better to show more about the consistency and inconsistency compared with the similar research that conducted in the same/different regions, and the possible reasons for the difference.

√ Response: In order to explain more about the consistency and inconsistency with other studies and the reasons for the differences, we have made an explanation in line 331-342 of the article.

Chadsuthi et al. [69] studied the influenza prediction model in central and southern Thailand, and found that the ARIMAX model including the average temperature lagging 4 months and the minimum relative humidity lagging 2 months was the model suitable for the central region, while the ARIMAX model including the minimum relative humidity lagging 4 months was the best model suitable for the southern region. In Abidjan, the inclusion of rainfall can improve the performance of influenza fitting and forecasting models [70]. Although most studies have found that the inclusion of meteorological factors can improve the performance of influenza prediction models, due to the differences in regional climatic conditions and human factors, the meteorological factors included in the optimal multivariate models for influenza prediction in each region are different. This is a further reminder of the regional nature of influenza prediction models.

 Why the plots in Fig 2 had two titles? Please remove one if possible. 

√ Response: We are very sorry for the misunderstanding caused by the question in Figure 2, but the two headings in Figure 2 are different contents. In order to avoid misunderstanding, we have modified Figure 2, as shown in Figure 2.

Fig 2 Prediction fitting of SARIMA(X) model on the number of influenza cases in Lanzhou from January 2017 to December 2017 (A represents the Univariate SARIMA model, B represents the Multivariate SARIMAX model)

 Some typos: e.g. Line176: 2016/012; Line 210: “Spearman line correlation”; Line 221: “。”; Table 4 please show the values with same number of decimals.

√ Response: We are very sorry for the spelling mistakes in the article. We have been revising the position of the article.

Line 176: the first 36 months (2014/01-2016/12) were used to construct the time series model

Line 210: Spearman correlation analysis was used to analyze the daily influenza cases

Line 221: (r = -0.719, P < 0.01).

We have unified the number of decimal places in Tables 5. However, in order to keep the consistency of P decimal places in Table 4, we still keep four decimal places of P。 in Table 4. After R software were run, AICs and BICs values only display two decimal places, so they cannot keep the same number of decimal places as P。.

Other changes:

 Line 3, Since the author of the revision of the article, Hai Zhu, has made a great contribution to the revision of this article, we have decided to work with Hai Zhu as a common author.

 Line 7, We added a second affiliation of Meixia Du, “Gansu Provincial Cancer Hospital, Gansu Lanzhou 730050, China”.

 Line 21, We added the notation for the contributed equally to this work, “¶ These authors contributed equally to this work”.

 Line 367, We changed the order of authors' contributions, “Hai Zhu: Methodology, Data Collection, Article modification”.

Special thanks to you for your good comments. 

We tried our best to improve the manuscript and made some changes in the manuscript. These changes will not influence the content and framework of the paper. And here we did not list the changes but marked in blue in revised paper.

We appreciate for Editors and Reviewers’ warm work earnestly, and hope that the correction will meet with approval.

Once again, thank you very much for your comments and suggestions.

Best regards,

Dr.Xiaochun Yin

School of Public Health

Gansu University of Chinese Medicine

---

## [Editor Report · Decision Letter 2]

19 Oct 2022

Exploration of influenza incidence prediction model based on meteorological factors in Lanzhou, China, 2014-2017

PONE-D-21-37259R2

Dear Dr. Yin,

We’re pleased to inform you that your manuscript has been judged scientifically suitable for publication and will be formally accepted for publication once it meets all outstanding technical requirements.

Kind regards,

Ka Chun Chong

Academic Editor

PLOS ONE

Additional Editor Comments (optional):

The authors have addressed well to all the comments. Congratulations.
---

## [Editor Report · Acceptance letter]

6 Dec 2022

PONE-D-21-37259R2 

Exploration of influenza incidence prediction model based on meteorological factors in Lanzhou, China, 2014-2017 

Dear Dr. Yin:

I'm pleased to inform you that your manuscript has been deemed suitable for publication in PLOS ONE. Congratulations! Your manuscript is now with our production department. 

Kind regards, 

on behalf of

Dr. Ka Chun Chong 

Academic Editor

PLOS ONE